

# An assessment of the accuracy of global rainfall estimates without ground-based observations

Christian Massari[1], Wade Crow[2], and Luca Brocca[1]

[1]Research Institute for Geo-Hydrological Protection, National Research Council, Perugia, Italy
[2]United States Department of Agriculture - Hydrology and Remote Sensing Laboratory, Beltsville, Maryland, USA

*Correspondence to:* Christian Massari (christian.massari@irpi.cnr.it)

**Abstract.** Satellite-based rainfall estimates have great potential value for a wide range of applications, but their validation is challenging due to the scarcity of ground-based observations of rainfall in many areas of the planet. Recent studies have suggested the use of Triple Collocation (TC) to characterize uncertainties associated with rainfall estimates by using three collocated products of this variable. However, TC requires the simultaneous availability of three products with mutually-uncorrelated errors, a requirement that is difficult to satisfy among current global precipitation datasets.

In this study, a recently-developed method for rainfall estimation from soil moisture observations, SM2RAIN, is demonstrated to facilitate the accurate application of TC within triplets containing two state-of-the art satellite rainfall estimates and a reanalysis product. The validity of different TC assumptions are indirectly tested via a high quality ground rainfall product over the Contiguous United States (CONUS), showing that SM2RAIN can provide a truly independent source of rainfall accumulation information which uniquely satisfies the assumptions underlying TC. On this basis, TC is applied with SM2RAIN on a global scale in an optimal configuration to calculate, for the first time, reliable global correlations (versus an unknown truth) of the aforementioned products without using a ground benchmark dataset.

The analysis is carried out during the period 2012-2015 using daily rainfall accumulation products obtained at $1° \times 1°$ spatial resolution. Results convey the relatively high accuracy of the satellite rainfall estimates in Eastern North and South America, South Africa, Southern and Eastern Asia, Eastern Australia as well as Southern Europe and complementary performances between the reanalysis product and SM2RAIN, with the first performing reasonably well in the northern hemisphere and the second providing very good performance in the southern hemisphere.

The methodology presented in this study can be used to identify the best rainfall product for hydrologic models with sparsely-gauged areas and provide the basis for an optimal integration among different rainfall products.

## 1 Introduction

Thanks to the combined use of microwave and infrared sensors, the quality of available satellite rainfall estimates has significantly increased in the few last decades. This strategy – also known as multi-sensor approach – has produced a number of different satellite rainfall products that either map infrared (IR) radiances to more direct passive microwave (PMW) retrievals (generally termed "blended" algorithms) or morph PMW rainfall using IR measurements (generally termed "morphing" algo-





rithms). The new Global Precipitation Measurement Mission (GPM, Hou et al., 2014) has successfully expanded the concept of multi-sensor integration. Trough the Integrated Multi-satelliE Retrievals for GPM (IMERG) algorithm, rainfall estimates from the various precipitation-relevant satellite PMW and IR missions are intercalibrated, merged and interpolated with the GPM Combined Core Instrument product to produce rainfall accumulation estimates with an unprecedented accuracy. Despite these

technical advancements, the precipitation community still struggles to show a clear picture of the actual increased accuracy of satellite rainfall estimates in many areas of the world because validation studies rely upon the availability of high-quality (and sufficiently dense) ground-based rainfall instrumentation (e.g. rain gage and radars).

Many studies (e.g., Ebert et al., 2007; Sapiano and Arkin, 2009; Tian et al., 2007; Stampoulis and Anagnostou, 2012) have investigated the error associated with remotely-sensed precipitation products by comparing their estimates with those

collected by ground-based observations assuming they represent the zero-error rainfall. However, the physical characteristics of precipitation, particularly at finer spatial and temporal resolutions, necessitate frequent, systematic and sufficiently dense validation measurements – requirements that are often not met within data-scarce regions of Africa, Asia and South America. Indeed, despite their relative accuracy, the distribution of available gauges significantly varies around the world. Much of the land masses (representing 25 – 30 % of the Earth's surface) have measurement networks, although those networks with good

gauge densities are limited (Kidd et al., 2017).

The current networks of surface observations are therefore often not sufficient for the quantitative assessment of the error associated with satellite rainfall estimates. Moreover, despite the relatively higher accuracy of rainfall estimates that can be obtained by rain gauges, they are not error-free. (Peterson et al., 1998; Villarini et al., 2008). Therefore, evaluating the performance of different satellite rainfall products with ground based observations is challenging due to the scarcity of such

observations and of the inherent error contained in their estimates.

Based on the work of Adler et al. (2009), Tian and Peters-Lidard (2010) estimated the uncertainties of satellite rainfall estimates by using the measurement spread of coincidental and collocated estimates from an ensemble of six different satellite-based datasets, thus providing a globally consistent methodology that does not require ground-based based validation data. The analysis yielded a lower bound estimate of the uncertainties, and a consistent global view of the error characteristics and their

regional and seasonal variations. However, the authors showed that the analysis is able to provide only a relative estimation of the measurement uncertainties because these data sets are not entirely independent measurements.

An alternative approach for assessing the quality of satellite rainfall products was proposed by Roebeling et al. (2012) and Alemohammad et al. (2015) based on the Triple Collocation (TC) method (Stoffelen, 1998). The first applications of TC concerned geophysical variables such as ocean wind speed and wave height (Stoffelen, 1998). More recently, it has been used

extensively to estimate errors in soil moisture (SM) products (Crow and Van Den Berg, 2010; Miralles et al., 2010; Dorigo et al., 2010; Draper et al., 2013; Su et al., 2014; Gruber et al., 2016). Given three estimates of the same variable, the main assumptions of the method are the: i) stationarity of the statistics, ii) linearity between the three estimates (versus the same target) across all time scales and iii) existence of uncorrelated error between the three estimates.

In the work of Roebeling et al. (2012), the authors determined the spatial and temporal error characteristics of three precipita-

tion datasets over Europe (a visible/near infrared, a weather radar and gridded rain gauge products) showing that it can provides





realistic error estimates. The authors ensured a Gaussian distribution of the error by averaging the dataset over a sufficiently long period (10 days) and re-gridding to sufficiently low spatial resolutions (0.25x0.25°). Alemohammad et al. (2015) applied TC method to 14 days cumulated rainfall estimates derived from satellite, gauge, radar and models in order to retrieve the error and the correlation of each dataset in United States. They also proposed the use of a logarithmic (i.e., multiplicative) error

model which almost certainty provides a more realistic description of rainfall accumulation errors at fine scale/time scales. In addition, they calculated the theoretical correlation of each product with the unknown truth by using Extended TC (ETC) (McColl et al., 2014) by analysing the covariance matrix of the three datasets.

TC can theoretically provide error and correlations of three products (a triplet) without using ground based observations – provided that each of the three products is afflicted by mutually-independent errors. However, given that state-of-the-art satellite

rainfall products use a highly-overlapping set of common sensors for the retrieval of rainfall (see section 2.1 for further details), there is an inherent difficulty in obtaining triplets with mutually-independent errors. Therefore, additional - highly independent - sources of rainfall accumulation estimates are needed.

Recently, Brocca et al. (2014) developed a method for estimating rainfall accumulation amounts directly from satellite SM observations based on the principle that the soil can be treated as a "natural raingauge". In contrast with classical satellite rainfall

products, this new bottom-up approach attempts to measure rainfall by calculating the difference between two successive SM measurements derived from a satellite SM product. In this respect, SM2RAIN offers a unique opportunity for applying the TC analysis because, being wholly independent from any other rainfall estimate, it can be used in place of a ground-based product. This opportunity has not yet been explored and could provide an appropriate basis for applying TC on a global scale without requiring the availability of ground-based rainfall accumulation data.

In this study, TC analysis is applied to the rainfall accumulation estimates derived from: 1) ERA-Interim (Dee et al., 2011), 2) SM2RAIN (Brocca et al., 2014) via inversion of Advanced SCATterometer (ASCAT, (Wagner et al., 1999) SM data, 3) the NOAA Climate Prediction Center morphing (CMORPH) (Joyce et al., 2004) and 4) the TRMM Multi-satellite Precipitation Analysis TMPA 3B42RT (Huffman et al., 2007) product over the CONtiguous United States (CONUS). An assessment of the reliability of subsequent TC results, is conducted by direct comparison with the analogous evaluation results obtained via direct

comparisons with the Climate Prediction Center (CPC) Unified Gauge-Based Analysis of Global Daily Precipitation (hereafter as CPC) product. These assessments will be carried out with and without the use of SM2RAIN rainfall accumulation products to isolate the value of SM-based rainfall estimates for the evaluation of global rainfall products.

The paper is organized as follows. Section 2 contains data and method descriptions; in particular, the products used for the analysis are described in section 2.1; the theoretical background for TC is in sections 2.2 and 2.2.1; the description of the

performance scores used for the evaluation of the results is discussed in section 2.3, and sections 2.4 and 2.5 describe SM2AIN and the experiment setup. Section 3 contains the results and discussion, and final remarks are contained in section 4.



## 2   Data and Methods

### 2.1   Rainfall and soil moisture products

#### 2.1.1   CPC

The 0.5°x0.5° gauge-based CPC product is used to evaluate the satellite-based rainfall estimates over CONUS and verify
evaluations provided by TC. Given the high raingauge density associated with this product across CONUS (Figure 1), along
with the common practice of using ground-based rainfall data to validate satellite-based rainfall retrievals (Huffman et al.,
1997), CPC is expected to provide a reasonable proxy of true rainfall accumulation over CONUS. This assumption will be
verified below. Figure 1 illustrates that the spatial density of CPC gauge coverage (calculated as average number of rain gauge
observations per day) during 2007-2012 is high in Eastern CONUS and along the western coast of CONUS but relatively
lower in many parts of the Central CONUS. CPC rainfall observations are aggregated to a 1°x1° spatial resolution by simple
averaging.

#### 2.1.2   ASCAT data

a ASCAT (Bartalis et al., 2007) is a real-aperture radar instrument on board the MetOp satellites that measures radar backscatter
at C-band (5.255 GHz) and VV polarization. It has a spatial resolution of 25 km (resampled at 12.5 km), and it is available since
2007. The surface SM product (equivalent to a depth of 2-–3 cm of the soil) is calculated from the backscatter measurements
through the time-series-based change detection approach described in Wagner et al. (1999). The SM is measured in relative
terms (degree of saturation) with respect to historical minimum and maximum values. Here, we used the ASCAT dataset
produced using the Soil Water Retrieval Package (WARP) (Naeimi et al., 2009) (v5.5) from Vienna University of Technology
(TU-Wien), provided as SM product H109 by the "EUMETSAT Satellite Application Facility on Support to Operational
Hydrology and Water Management (H-SAF)". ASCAT data are used solely for rainfall estimation with SM2RAIN. For further
details, the reader is referred to section 2.4.

#### 2.1.3   TMPA 3B42RT

TMPA 3B42RT, version 7 (http://trmm.gsfc.nasa.gov), combines rainfall estimates from various satellite sensors. The mul-
tisatellite platform uses the TRMM Microwave Imager (TMI) on board of TRMM satellite, the Special Sensor Microwave
Imager (SSM/I) on board the Defense Meteorological Satellite Program (DMSP) satellites, the Advanced Microwave Scan-
ning Radiometer for Earth observing system (AMSRE) on board the National Aeronautic and Space Administration (NASA)
AQUA satellite, the Advanced Microwave Sounding Unit-B (AMSU-B) on board the National Oceanic and Atmospheric Ad-
ministration (NOAA) satellite series and GEO IR rainfall estimates . The TMPA 3B42RT estimates are produced in three steps:
1) the PMW estimates are calibrated with sensor-specific versions of the Goddard Profiling Algorithm (GPROF; Kummerow
et al., 1996) and combined, 2) IR rainfall estimates are created using the PMW estimates for calibration, and 3) PMW and
IR estimates are then combined. The 3B42RT product is provided by NASA with a temporal resolution of 3 h and a spatial





resolution of 0.25°. The cumulated daily rainfall, available from March 2000, is obtained by simply summing the eight 3-h time windows for each day. The global coverage of the product is +50°/-50° latitude. To match the CPC spatial resolution, collocated TMPA 3B42RT estimates are aggregated to 1 ° spatial resolution by simple averaging.

### 2.1.4 CMORPH

CMORPH uses a Lagrangian approach to construct high-resolution global precipitation maps from the satellite IR and PMW observations (Joyce et al., 2004). This technique uses precipitation estimates that have been derived from PMW observations exclusively, and whose features are transported via spatial propagation information which is obtained entirely from IR data. It incorporates precipitation estimates derived from the PMW on board the NOAA satellites (SSM/I, the NOAA satellite series and AMSU-B) as well as AMSR-E and TMI aboard NASA's Aqua and TRMM spacecraft, respectively. Precipitation

estimates are obtained as follows. First, advection vectors of cloud and precipitation systems are computed using consecutive geostationary IR images in 30 min intervals. These advection vectors are then applied to propagate the precipitating cloud systems observed by the PMW measurements along the advection vectors in both forward and backward directions toward the target time of the precipitation analysis. The final precipitation analysis value at a grid box is defined as the weighted mean of the estimates from the forward and backward propagations with the weights inversely proportional to the time separation

between the target analysis time and the PMW observations. In this study, we used the daily (derived from 3-hourly aggregation) estimates of precipitation at 0.25 ° latitude/longitude resolution, distributed over the globe (+60°/-60° of latitude) by the NOAA Center for Weather and Climate Prediction. Note that the CMORPH version used in this study is the raw version which does not uses gauge information. To match the CPC spatial resolution, collocated CMORPH estimates are aggregated to 1 ° spatial resolution.

### 2.1.5 ERA-Interim

The European Centre for Median-range Weather Forecasts (ECMWF) produces the ERA-Interim atmospheric, ocean and land reanalysis . ERA-Interim provides medium-range global forecasts for some environmental variables that include soil temperature, evaporation, SM and rainfall. Products are available from 1th January 1979 to now. The forecast model incorporated in the ERA-Interim reanalysis is based on the ECMWF Integrated Forecast System (Cy31r2) forecast model (Dee

et al., 2011), with a spectral horizontal resolution of about 80 km and 60 vertical levels. The ERA-Interim forecast precipitation is the sum of two components which are computed separately in the model: large-scale stratiform precipitation (Tompkins et al., 2007) and smaller scale precipitation which originates solely from the parameterization of convection (Bechtold et al., 2004). Further information can be found at the ECMWF website (http://www.ecmwf.int). In this study, daily precipitation values are obtained from the temporal aggregation of ERA-Interim 12-hourly precipitation accumulation estimates

(http://apps.ecmwf.int/datasets/) while co-location with CPC observations is determined by the nearest-neighbour method.



## 2.2 Triple Collocation analysis: general concepts

Here we apply the method of (McColl et al., 2014) to robustly estimate the correlation of a particular rainfall measurement system with the truth. Suppose we have three systems $X_i$, measuring the true variable $t$ and afflicted by additive random error

$$X_i = X_i' + \varepsilon_i = \alpha_i + \beta_i t + \varepsilon_i \tag{1}$$

where $X_i$ ($i =$ 1, 2, 3) are collocated measurement systems linearly related to the true underlying value $t$ with additive random

5 errors $\varepsilon_i$, and $\alpha_i$ and $\beta_i$ are the ordinary least squares intercepts and slopes. Assuming that the errors from each system have zero mean ($E(\varepsilon_i)$=0) and are mutually uncorrelated ($Cov(\varepsilon_i, \varepsilon_j)$ =0, with $i \neq j$) and orthogonal with respect to $t$ ($Cov(\varepsilon_i, t)$=0), the covariance between $X_i$ is:

$$Q_{ij} = Cov(X_i, X_j) = \begin{cases} \beta_i \beta_j \sigma_t^2, & \text{for } i \neq j \\ \\ \beta_i^2 \sigma_t^2 + \sigma_{\varepsilon_i}^2, & \text{for } i = j \end{cases} \tag{2}$$

By defining the new variable $\theta_i = \beta_i \sigma_t$, known as the sensitivity of the variable $X_i$, Eq. (2) becomes:

$$Q_{ij} = \begin{cases} \theta_i \theta_j, & i \neq j \\ \\ \theta_i^2 + \sigma_{\varepsilon_i^2}, & \text{for } i = j \end{cases} \tag{3}$$

which is a system of six equations in six unknowns from which we derive (McColl et al., 2014):

$$\sigma_\varepsilon = \begin{bmatrix} \sqrt{Q_{11} - \frac{Q_{12}Q_{13}}{Q_{23}}} \\ \sqrt{Q_{22} - \frac{Q_{12}Q_{23}}{Q_{13}}} \\ \sqrt{Q_{33} - \frac{Q_{12}Q_{23}}{Q_{12}}} \end{bmatrix} \tag{4}$$

10    From Eq. (2), using the definition of the correlation and covariance we can write:

$$\theta_i = \rho_{t,X_i} \sqrt{Q_{ii}} \tag{5}$$

where $\rho_{t,X_i}$ is the correlation coefficient between $t$ and $X_i$. Since $\sqrt{Q_i i}$ is already estimated from the data, and we can solve for $\theta_i$ using Eq. (4), $\rho_{t,X_i}$ (McColl et al., 2014):

$$\rho_{\mathbf{t},\mathbf{X}} = \pm \begin{bmatrix} \sqrt{\frac{Q_{12}Q_{13}}{Q_{11}Q_{23}}} \\ sign(Q_{13}Q_{23})\sqrt{\frac{Q_{12}Q_{23}}{Q_{13}Q_{22}}} \\ sign(Q_{12}Q_{23})\sqrt{\frac{Q_{13}Q_{23}}{Q_{12}Q_{33}}} \end{bmatrix} \tag{6}$$

which provides the temporal correlation of each product with the unknown truth. Hereafter, when talking about $\rho_{t,X_i}$ or its squared value $\rho_{t,X_i}{}^2$, we will refer to the correlation of the product $X_i$ with the unknown truth. $\rho$ will be also used to refer to

15  this variable but in more general terms.



### 2.2.1 Some considerations about the application of the Triple Collocation to rainfall estimates

It is generally accepted that a multiplicative model is more appropriate for describing errors in rainfall estimates (Hossain and Anagnostou, 2006). Based on this assumption, (Alemohammad et al., 2015) proposed the application of TC to the rainfall by introducing a multiplicative error model:

$$R_i = a_i T^{\beta_i} e^{\varepsilon_i} \tag{7}$$

in which $R$ is the rainfall intensity estimate from product $i$, $T$ is the true rainfall intensity and $a_i$ is the multiplicative error. By transforming Eq. (7) in the log-space we obtain an equation equivalent to Eq. (1) where $X = log(R)$, $t = log(T)$ and $\alpha_i = log(a_i)$. In this way, the development of TC expressed in Eqs. 2–6, can be applied to the - potentially more relevant - case of multiplicative rainfall accumulation errors. The resulting log-RMSE can then be back-transformed into linear rainfall accumulation errors by exploiting a Taylor series expansion of the logarithm (see Alemohammad et al., 2015 for further details).

The main difficulty of this approach is its inability to consider the presence of zero values in the rainfall time series. To reduce their presence, Alemohammad et al. (2015) considered biweekly rainfall estimates and simply removed remaining zeros in this time series. This has two implications. First, the biweekly rainfall error may differ from the error of a shorter accumulation period (e.g. daily) because the daily signal has a substantially different character with respect to the biweekly one due to the higher presence of zero values. Second, the method may not be appropriate in very dry climates where even biweekly values of rainfall can contain a significant number of zero accumulation values.

For the reasons mentioned above, we apply TC in two different ways: i) to the rainfall time series using an additive error model, and ii) to log-transformed rainfall estimates using the multiplicative error model (by first removing rainfall accumulation values equal to zero). Comparisons of these two different approaches will provide insights regarding the appropriateness of various error model assumptions for rainfall estimates at a daily accumulation time scale.

### 2.3 Performance scores

In section 2.2, it has been demonstrated that TC can provide both error variances and correlation against an unknown truth for three collocated estimates of the same variable. When dealing with error variances, the products have to be rescaled to a common reference data space. However, such a rescaling imposes spatial patterns within the derived error metric which reflects the climatology of the chosen reference (Gruber et al., 2016). To this end, McColl et al. (2014) noted that correlation coefficients can provide important new information about the performance of the measurement systems with respect to the absolute error variances obtained via Eq. (4) with the added advantage of not requiring the arbitrary definition of one system as a scaling reference. Indeed, $\rho^2$ represents the unbiased signal to noise ratio, scaled between 0 and 1, which provides a measure of the relative similarity between two signals, independently from their phase differences. This was also underlined by Gruber et al. (2016), who showed that $\rho^2$ is the complement of the $fRMSE = \sigma_\varepsilon^2/\sigma^2$ introduced by (Draper et al., 2013) ($\rho^2 = 1 - fRMSE$) which was used previously to remove the dependency of the error variance pattern on the spatial climatology of the chosen reference. Gruber et al. (2016) also pointed out that the absolute error variance provides only limited information





about the true dataset quality because a certain amount of noise can be either acceptable or unacceptable depending on the strength of the underlying signal (i.e., its variance). Therefore, we focus here only on $\rho^2$, or analogously, on its root square $\rho$, i.e., Eq. (6).

 As discussed above, a key goal is determining the relative accuracy of TC correlations obtained with and without the use
of SM2RAIN-based rainfall accumulation products. Assuming that $R_{X_i}$ (or simply $R$) is the Pearson correlation coefficient between the product $X_i$ and CPC, the main question is: how accurately can (TC-based) $\rho_{t,X_i}$, which utilize no ground observations, reproduce spatial patterns of (CPC-based) $R_{X_i}$? We should expect a bias between the two (i.e., $R_{X_i} \leq \rho_{t,X_i}$) because – while relatively accurate – CPC still contains representativeness errors (due to limitations in raingauge density) and measurement errors due to wind and instrument inaccuracies, in contrast Eq.(6) provides the correlations with an error-free truth.
Nevertheless, if the TC hypothesis holds, the relative rank between the products predicted by TC should accurately reflect that obtained via direct comparisons with ground observations.

 In order to evaluate the similarity between correlation-based maps of $\rho_{t,X_i}$ and $R_{X_i}$ a spatial correlation index $SC$ was calculated as the spatial Pearson correlation coefficient between maps of $R_{X_i}$ and $\rho_{t,X_i}$. The closer $SC$ is to one, the more similar the two maps are and the more satisfied the assumptions of TC. In addition, based on the values of $\rho_{t,X_i}$ and $R_{X_i}$, we
are able to sort the products according to their relative performance for each pixel in the analysis. That is, considering three products $X_i$, the rank value $RK_i$ to be assigned to each product $i$ will be 1 if $\rho_{t,X_i}$ is the highest, 3 if it is the lowest and 2 if it is comprised between the minimum and the maximum value. If the same is done with $R_{X_i}$, the consistency of the resulting maps for each product will provide feedback regarding the validity of assumptions underlying the application of TC. For the quantification of the discrete maps, we also calculate the number of pixel providing equal sorting of the products obtained
based on $R_{X_i}$ and $\rho_{t,X_i}$.

## 2.4 SM2RAIN and its application to ASCAT data

SM2RAIN (Brocca et al., 2014) is a method of rainfall estimation which uses two successive SM retrievals (i.e., at the current and at a past time step) to estimate the rainfall accumulated within the time interval between the two retrievals. It exploits the soil water balance equation with appropriate simplifications valid only for liquid precipitation (Tian et al., 2014):

$$Z^* ds(\tau)/d\tau = p(\tau) - r(\tau) - e(\tau) - g(\tau) \qquad (8)$$

where $Z^*$ is the soil water capacity (soil depth times soil porosity); $s(\tau)$ is the relative saturation of the soil or relative SM; $\tau$ is the time and $p(\tau)$, $r(\tau)$, $e(\tau)$ and $g(\tau)$ are the rainfall, surface runoff, evapotranspiration and drainage rates, respectively. Under unsaturated soil conditions, and assuming negligible evapotranspiration rate during rainfall and Dunnian runoff, solving Eq. (8) for rainfall yields:

$$p(\tau) = Z^* ds(\tau)/d\tau + as(\tau)^b \qquad (9)$$

 Note that in Eq. (9) the drainage rate has been expressed with a power law function of the type $g = as^b$ (Famiglietti and
Wood, 1994), where $a$ and $b$ two model parameters. When the soil is fully saturated, no rainfall can be estimated from SM; however, at the scale of satellite pixel, the soil is rarely saturated (except in some exceptional places like tropical forests).





The SM2RAIN parameters $a$, $b$ and $Z^*$ can be estimated either by using a rainfall dataset as a reference or assigned based on soil properties. In this study, in order to maximize the independence of SM2RAIN predictions, SM2RAIN parameters were not calibrated and were instead assumed constant in space as in Koster et al. (2016). In particular, the drainage rate (the second term in Eq. (9)) was assumed linearly related with SM ($b = 1$) and $a = 3.7$ mm/day and $Z^* = 62$ mm based on results obtained in previous studies (Brocca et al., 2014). Note that, $Z^*$ does not have a significant influence on the results because we are using a correlation based metric. In addition, it should be noted that, while maximizing the independence of SM2RAIN rainfall accumulation estimates, the use of this default calibration approach results in sub-optimal SM2RAIN performance. Superior SM2RAIN can easily be obtainable via calibration against existing satellite rainfall accumulation products.

Daily rainfall estimates from SM2RAIN were obtained by using linearly-interpolated (at at 00:00 UTC) ASCAT data with a maximum allowable data gap of 5 days. The obtained 0.25 °x0.25 ° rainfall estimates were then aggregated to the 1 °x1° spatial resolution by simple averaging of the collocated pixels with CPC. Hereinafter, the thus obtained product is referred to as SM2RAIN for simplicity.

## 2.5 Experimental setup

A TC analysis was carried out using five different daily rainfall accumulation triplets: 1) ERA-Interim-SM2RAIN-3B42RT (Triplet A in the following) 2) ERA-Interim-SM2RAIN-CMORPH (Triplet B) 3) ERA-Interim-3B42RT-CMORPH (Triplet C), 4) ERA-Interim-3B42RT-CPC (Triplet D) and 5) ERA-Interim-CMORPH-CPC (Triplet E). Triplets A and B serve to assess the ability of SM2RAIN to provide meaningful TC results. Triplet C provides an alternative to triplets A and B which contains two rainfall satellite products (with potentially cross-correlated errors). Triplets D and E serve only to evaluate the general performance of the CPC product (within CONUS) and to provide alternative triplets to A and B which do have utilize SM2RAIN. As a result, they will only be used for initial considerations about TC robustness and to evaluate the relative quality of the CPC product. Triplets A, B and C will be then used in the reminder of the paper to demonstrate the potential utility of SM2RAIN.

The analysis was carried out first across CONUS and then on a global scale using only ERA-Interim, 3B42RT, CMORPH and SM2RAIN during the period 2012-2015. Over CONUS it was confirmed that the sample size to apply TC was sufficient within the entire study domain (Gruber et al., 2016) while for the global analysis, grid cells with inadequate sample size were individually masked out of the analysis. The Extended TC analysis was applied for both additive and multiplicative error model assumptions. For the latter, we first removed zeros (they are about 80% of the rainfall values that means about 450 values of non null rainfall) values from the signal and then applied a log-transformation to the remaining daily rainfall estimates. This reduction in sample size may affect TC results by making the analysis with log-precipitation estimates statistically less robust.

## 3   Results and discussion

In this section, we present the results obtained from the application of TC (for both additive and multiplicative error models) by following the subsequent methodological steps: 1) calculating TC based correlations ($\rho_{t,X_i}$) for Triplets A, B, C, D and





E over CONUS and providing an assessment of the CPC product (section 3.1); 2) understanding the adequacy of TC results based on the spatial similarity between (TC-based) $\rho_{t,X_i}$ and (CPC-based) $R_{X_i}$ (along with their relative rank) over CONUS in order to identify the optimal configuration for applying TC, and, 3) applying the optimal-configured TC on a global scale to calculate $\rho_{t,X_i}$ globally for the selected rainfall products (section 3.3).

## 3.1  Assessment of CPC accuracy

As described above, our first goal is assessing the relative performance of the CPC product. Table 1 shows mean $\rho_{t,X_i}$ (obtained via the spatial average of 0.25-degree CONUS grid-cells). Regardless of the triplet or error model applied, the TC analysis summarized in Table 1 indicates that CPC is the most accurate product (mean TC-based correlation close to 0.9 for the additive error model and close to 0.8 for the multiplicative one) which strengthens our assumption that within CONUS, CPC can be used as a benchmark to evaluate the optimal TC configuration for rainfall product evaluation. In addition, its correlation spatial pattern (not shown) provides very good performance almost everywhere except in the Central US where the spatial density of available rain gauges shown in Figure 1 is relatively lower. Based on this, in the next section we will consider the CPC product as an appropriate benchmark for the selection of an optimal TC configuration which does not utilize a gauge-based precipitation product (and is therefore potentially applicable at a global scale).

## 3.2  Optimal Triple Collocation configuration

Figures 2 (a, b, c and d) plot CPC-based Pearson correlation coefficients (i.e., $R_{X_i}$) for ERA-Interim, 3B42RT, CMORPH and SM2RAIN obtained with the assumption of additive error model (for multiplicative error model results the reader is referred to Figure S1 of the supplementary material). A comparison of these results with TC-based correlations (i.e., $\rho_{t,X_i}$) shows that $\rho_{t,X_i}$ ae biased high with respect to $R_{X_i}$. This is expected given that CPC is not free of errors whereas TC should theoretically provides the correlation with respect to the truth.

The spatial agreement between $\rho_{t,X_i}$ and $R_{X_i}$ is examined in Table 2 and Figure 2. In particular, Figure 2 shows that Triplets A (e, f, g in the figure) and B (h, i, l) accurately reproduce CPC-based results plotted in Figure 2 (a, b, c and d), although they are characterized by higher values as underlined above (see section 2.3 for further details). This similarity is higher in Eastern and Western US and lower in the Central US especially for ERA-Interim and SM2RAIN. This lower agreement in the Central US is likely due to the lower rain gauge density of CPC here (see Figure 1) which degrades the quality of the CPC product as benchmark. However, in contrast, TC-results based on Triplet C predicts a substantial different behaviour with correlation patterns which differ substantially relative to CPC-based benchmark results in Figure 2 (a, b, c and d). This suggests those triplets not containing SM2RAIN provide unreliable results. In particular, the simultaneous use of two satellite-based rainfall products in Triplet C leads to an overly-optimistic assessment of their accuracy. This is likely due to cross-correlated errors in 3B42RT and CMORPH rainfall accumulation products which causes TC to misinterpret their mutually-consistency as an indication of high-accuracy (Yilmaz and Crow, 2014).

It is often important to understand which is the best available rainfall product provides the highest relative accuracy. As described in section 2.3, we ranked the products based upon how well they compare relative to each other using both $R$ and





$\rho$, respectively. Figure 3 shows the distribution – three products at time (d, e, f; k, l, m; r, s, t) – of the relative rank based on comparisons with the (CPC-based) $R_{X_i}$ of each triplet, while subplots (a, b, c; g, h, i; n, o, p) of the same figure, provide similar information except that the relative rank is based on TC (i.e., $\rho$). The latter shows a very similar pattern with respect to CPC-based rank for Triplets A and B; however, Triplet C yields again a distinct pattern with ERA-Interim being the worst

product and 3B42RT and CMORPH providing complementary performances. As in the comparisons discussed in Figure 2, this implies that triplets containing SM2RAIN (i.e., Triplets A and B) provide more robust evaluation information than triplets utilizing both 3B42RT and CMORPH together.

The same analysis carried out with the assumption of multiplicative error model (see Figure S2 in the supplementary material), shows similar findings but larger differences between the spatial distribution of the rank obtained with CPC and the

one with TC, especially for Triplet B. To quantity this agreement, we have calculated the percentage of pixels which are ranked the same in both TC- band CPC-results. (% of rank identified in Table 2). The table confirms the patterns observed in Figure 3 and Figure S2 of the supplementary material with Triplets A and B yielding the highest percentage of pixels with a common rank – ranging from 65 to 81 % for the additive error model, and 48 to 71 % for the multiplicative error model. As discussed above, worst results are obtained in both cases for Triplet C (percentage of correct ranking between 5% and 60%).

A quantification of the agreement between the spatial variations of the correlations both for additive and multiplicative error models was also derived by the use of the spatial correlation $SC$ in Table 2. The table shows that for Triplets A and B, when TC is used with the assumption of additive error model, $SC$ is relatively high with values ranging from 0.61 to 0.84 while for Triplet C provides substantially lower $SC$ for 3B42RT and CMORPH. A slightly different situation can be observed for the multiplicative error model. Here, $SC$ values are generally lower than those obtained by TC with the assumption of additive

error model. In particular, ERA-Interim provides the worst score. This is not clearly evident in the spatial distribution of $R$ and $\rho$ (see Figure S1 in the supplementary material for further details) which show some similarities at least for Triplets A and B.

In summary, the application of TC to the different triplets shows that:

1. CPC product performs relatively well over CONUS with a TC-derived correlation versus truth of 0.9 (assuming an additive error model) demonstrating its relatively high quality here and supporting its application as a benchmark data

set within CONUS.

2. TC-based correlations are similar among the triplets except for Triplet C (i.e, ERA-Interim, 3B42RT and CMORPH). This is likely due to existence of non-negligible cross-correlated errors between 3B42RT and CMORPH.

3. A comparison between $\rho_{t,X_i}$ and $R_{X_i}$ shows that $\rho_{t,X_i}$ are biased with higher values with respect to $R_{X_i}$. In addition, the pattern of $\rho_{t,X_i}$ and $R_{X_i}$ is similar for all triplets except for Triplet C, which shows inconsistencies relative to the CPC

benchmark for both the additive and multiplicative error model assumptions. The agreement, measured in term of spatial correlation (Table 2), provides higher scores for the additive error model assumptions with respect to the multiplicative one. This is likely due to a reduction of sampling power associated with the removal of daily rainfall accumulations equal to zero which are not acceptable in the log-transformation process. Therefore, it is possible that the observed differences in TC performance may shrink for larger sample sizes.





4. Retrieved spatial patterns of $\rho_{t,X_i}$ for the triplets containing SM2RAIN (Figure 2) shows a higher degree of similarity with (CPC-based) $R_{X_i}$ when we assume an additive (versus multiplicative) error model for daily rainfall accumulations.

On this basis, we can conclude that: i) TC results are unreliable unless SM2RAIN is used in the triplets and ii) the assumption of multiplicative error model in the application of TC at a daily time scale does not appear necessary.

## 3.3 Application of optimized TC approach

Based on the superior performance for Triplets A and B under the assumption of additive error model, we will apply this particular TC configuration approach to assess the accuracy (in terms of $\rho$) of daily rainfall accumulation estimates derived from 3B42RT, CMORPH, SM2RAIN and ERA-Interim first over CONUS (section 3.3.1 and Figure 2) and then on a global scale (section 3.3.2 and Figure 4).

### 3.3.1 CONUS

Over CONUS, ERA-Interim shows relatively better performance in Western and Eastern United States (US) with respect to the Central US where SM2RAIN is slightly superior. 3B42RT and CMORPH perform reasonably well in Eastern and along the coast of Western US while show worse performance in the Central US both in the south and in the north. On the contrary, SM2RAIN performs worse in Northern US probably due to the lower accuracy of the ASCAT data at high latitudes. The spatial pattern of these correlations is similar to the one in Gottschalck et al. (2005) and Ebert et al. (2007) who showed a general lower level of correlation of satellite-only rainfall products in the Central US due to effect of snow cover and frozen surface conditions. They are also similar to the results obtained by Alemohammad et al. (2015) using TC, who found a similar pattern of correlation of 3B42RT in a box covering a large part of southeastern US (however, the authors here assumed a multiplicative error model and biweekly rainfall accumulation estimates).

### 3.3.2 Global

On a global scale, 3B42RT (Figure 4a) shows relatively good performances in Eastern and Central South America, Southern and Central Africa, Southern and Eastern Asia, Eastern Australia, and Southern Europe while it performs relatively worse in Central Asia, Western Australia and in the southern part of the Sahel. The performance of CMORPH (Figure 4b) is similar to 3B42RT with slightly lower correlations in Australia, in the Horn of Africa and in Southern Asia. SM2RAIN (Figure 4c) performs reasonably well in Africa (except in the tropical forest), Australia, Mexico, Eastern South America and India and generally in the southern hemisphere while worse results are obtained in the northern hemisphere, in the tropical forests and at high latitudes. On the contrary, ERA-Interim (Figure 4d) provides much better results in the northern hemisphere with respect to the south of the planet (e.g., South America and South Africa) and performs relatively poor in Central and Northern Africa as well as in the tropical forests.

The results for 3B42RT and SM2RAIN are similar to those obtained in Brocca et al. (2014) who calculated the Pearson correlation coefficient with the Global Precipitation Climatology Center (GPCC, (Schamm et al., 2014)) dataset. Similar findings





are also present in Yong et al. (2015) (Table 2 of their paper) who compared different versions of the 3B42RT product against global CPC observations in US, East Asia, Europe and Australia. In their paper, the best results were obtained in Australia and in East Asia (Europe showed slightly lower performance) while lower performances were obtained in US like in our analysis. Further comparisons can be also considered with the recent work of Beck et al. (2017) who, in attempting to create a high-quality rainfall product specifically tailored for hydrological modelling, compared different satellite and modelled products globally with the Global Historical Climatology Network-Daily (GHCN-D, Menne et al., 2012) database. Their results (in terms of spatial pattern of correlation) are consistent with the ones obtained in our study over the US, East Asia and Middle East for CMORPH and 3B42RT, while less agreement is observed in Australia. For ERA-Interim, the results agree with our study in US, Europe and generally are better in the northern hemisphere, whereas they show some differences with SM2RAIN results in Australia, Africa and in South America, although in these areas a low number of available rain gauges cannot provide a clear picture of the real performance of the analysed products. Substantial differences between our study and the studies of Beck et al. (2017) and Yong et al. (2015) can likely be attributed to the quality of the benchmark dataset used for the evaluations. This is the main limitation of rainfall validation studies relying upon ground based observations for assessment. With the proposed TC-based approach, this issue can be overcome because ground observations are no longer required.

An interesting feature of the global evaluation of the products (Figures 4a and 4b, 4c and 4d), but also over CONUS between 3B42RT (or CMORPH) and SM2RAIN (Figure 2 triplets A and B), is the complementary of the performance among the products. Especially for Figures 4c and 4d, it can be seen that ERA-Interim performs very well in the northern hemisphere and worse in the southern hemisphere whereas SM2RAIN is relatively good in the south and worse in the northern hemisphere. Similar findings can be seen between the two state-of-the art satellite rainfall products (i.e., 3B42RT and CMORPH) and SM2RAIN over CONUS with the first performing better in Eastern US and the second in the Central and Western US. This opens up new possibilities for the integration of the products for obtaining a higher-quality merged rainfall estimate as also underlined in Ciabatta et al. (2015) and in Beck et al. (2017).

## 4 Summary and conclusions

The assessment of the performance of satellite rainfall products on a global scale is challenging due to significant limitations in the spatial coverage of high-quality, ground-based rain gauge observations. Provided that its underlying assumption are respected (see section 2.2), TC provides an alternative approach for evaluating global rainfall products without reliance on ground-based observations. Here, we describe how a new method for rainfall estimation based on SM observations (i.e., SM2RAIN) provides a rainfall product that is uniquely suited to satisfy the error independent assumptions at the heart of the TC approach.

The extended version of TC introduced by McColl et al. (2014) is able to provide the correlation with the (unknown) truth for each of the products applied within a particular triplet. To assess the robustness of correlated-based results obtained with TC, we used an area characterized by a high quality rainfall product (CPC dataset over CONUS, see Figure 1) with the assumption that it represents a good proxy of the true rainfall field. Therefore, if TC assumptions hold, Pearson correlation coefficients



computed against CPC should match those of TC – at least in terms of their relative values. Since we have two different error model options (i.e., additive and multiplicative) for the application of TC to rainfall data, we explored both.

Results demonstrate that daily rainfall accumulations provided by the CPC product are indeed relatively high quality compared to competing products (Table 1), thus supporting the assumption that it provides an acceptable proxy of the true rainfall field. Once it is established as a credible benchmark, CPC is used to evaluate 1) what type of triplets can be considered for a robust application of TC?, and 2) which model error assumption can be considered more appropriate? Triplets containing SM2RAIN and assuming an additive error model (Table 2) appear to provide the most robust TC results. Based on this, an optimal TC configuration was applied (for the first time) to globally evaluate daily rainfall accumulation derived from the 3B42RT and CMORPH, ERA-Interim and SM2RAIN products (Figures 4a and 4b, 4c and 4d) without the use of any ground-based data. Results convey the relatively high accuracy of daily rainfall accumulations derived from the satellite rainfall products (i.e., 3B42RT and CMORPH) in Eastern North and South America, South Africa, Southern and Eastern Asia, Eastern Australia as well as Southern Europe and complementary performances between ERA-Interim and SM2RAIN, with the first performing reasonably well in the northern hemisphere and the second providing very good performance in the southern hemisphere.

Based on the results obtained, we can therefore conclude that:

1. Despite the abundance of satellite rainfall estimates, their relative dependency impedes their use within the same triplet for the TC analysis, thus alternative independent products must be used for obtaining meaningful TC results. In particular, the use of two remotely-sensed rainfall products in a single triplet entails significant risk of a biased TC analysis.

2. Wholly independent daily rainfall accumulation products obtained from SM2RAIN are uniquely valuable for obtaining robust global evaluation statistics in absence of ground-based gauge observations. This is not only important for simple validation purposes but also for hydrological studies and applications within developing countries where ground-based rain gauge networks are often limited or even totally absent and an alternative product has to be chosen.

3. At the time/space scales examined here, the assumption of additive error model provides reasonable and robust results and no advantage is observed for a log-transformation of the time series (which allows for the consideration of a multiplicative error model). However, this result is likely to be scale-dependent and implies at the time/scale resolution of this analysis is sufficiently coarse such that averaging produces approximate additive/gaussian distributions (via the central limit theorem). Therefore, different results may be obtained at finer time/scales.

4. Both state-of-the art satellite rainfall estimates (i.e., 3B42RT and CMORPH) and SM-based rainfall estimates (i.e., SM2RAIN) performances are affected by the presence of snow cover and frozen soil conditions – thus these rainfall estimates may be unreliable at high latitudes and in mountainous regions. In these areas, a reanalysis product (i.e., ERA30 Interim) provides higher-quality rainfall estimates and should be considered in place of satellite-based estimates. SM-based rainfall estimates also work reasonably well in semi-arid climates (e.g., Sahel, Central Australia and Mexico) where the state-of-the-art satellite products report problems due to sub-cloud evaporation of hydrometeors (Ebert et al., 2007). Conversely, in wet climates (e.g, tropical forests) 3B42RT and CMORPH seem to be the only reliable option given that neither SM2RAIN nor ERA-Interim provide reasonable results.



5. Given the existence of complementary performances among the products, TC can potentially be a valuable tool for the characterization of their relative performances so as to be used for data fusion and assimilation experiments for obtaining more accurate rainfall estimates.

The question whether this analysis is valid for different spatio-temporal scales remain to be addressed and will be addressed in future studies. Also, removing zeros for obtaining log-transformed rainfall may not be ideal for testing the validity of the model error assumptions because it shortens the sample size, thus providing less robust TC results. Other strategies should be considered.

*Acknowledgements.* This Research was supported by the Short Term Mobility Program of the National Research Council of Italy and by the United States Department of Agriculture (USDA) and the NASA Terrestrial Hydrology Program award 13-THP13-0022 entitled "Integrating Satellite-based Surface Soil Moisture and Rainfall Accumulation Products for Improved Hydrologic Modeling". The first author wants to be grateful to Dr. Thomas Holmes and Dr. Concha Arroyo for their support and kindness during his period at the USDA-ARS Hydrology and Remote Sensing Laboratory (HRSL) in Beltsville (MD). We also would like to thank Dr. Luca Ciabbatta for processing part of the data used in the manuscript.





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



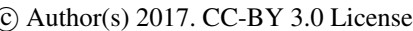

**Figure 1.** CPC gauge coverage during 2007-2012 expressed as average number of working raingauge per day within each 0.25-degree spatial grid cell.

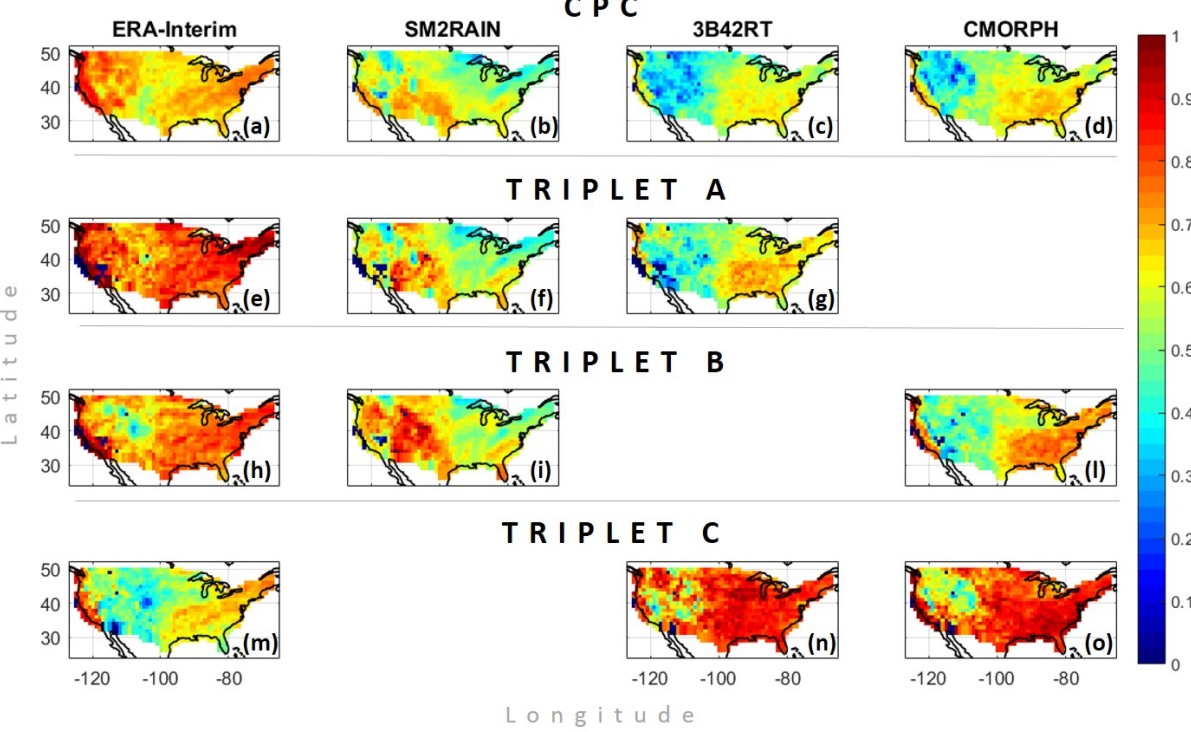

**Figure 2.** CPC-based (a, b ,c and d) and TC-based (e-o) correlation coefficient obtained for the triplets: i) ERA-Interim-SM2RAIN-3B42RT (e, f, g), ii) ERA-Interim-SM2RAIN-CMORPH (h, i, l) and iii) ERA-Interim-3B42RT-CMORPH (m, n, o) during the period 2012-2015 using an additive error model.





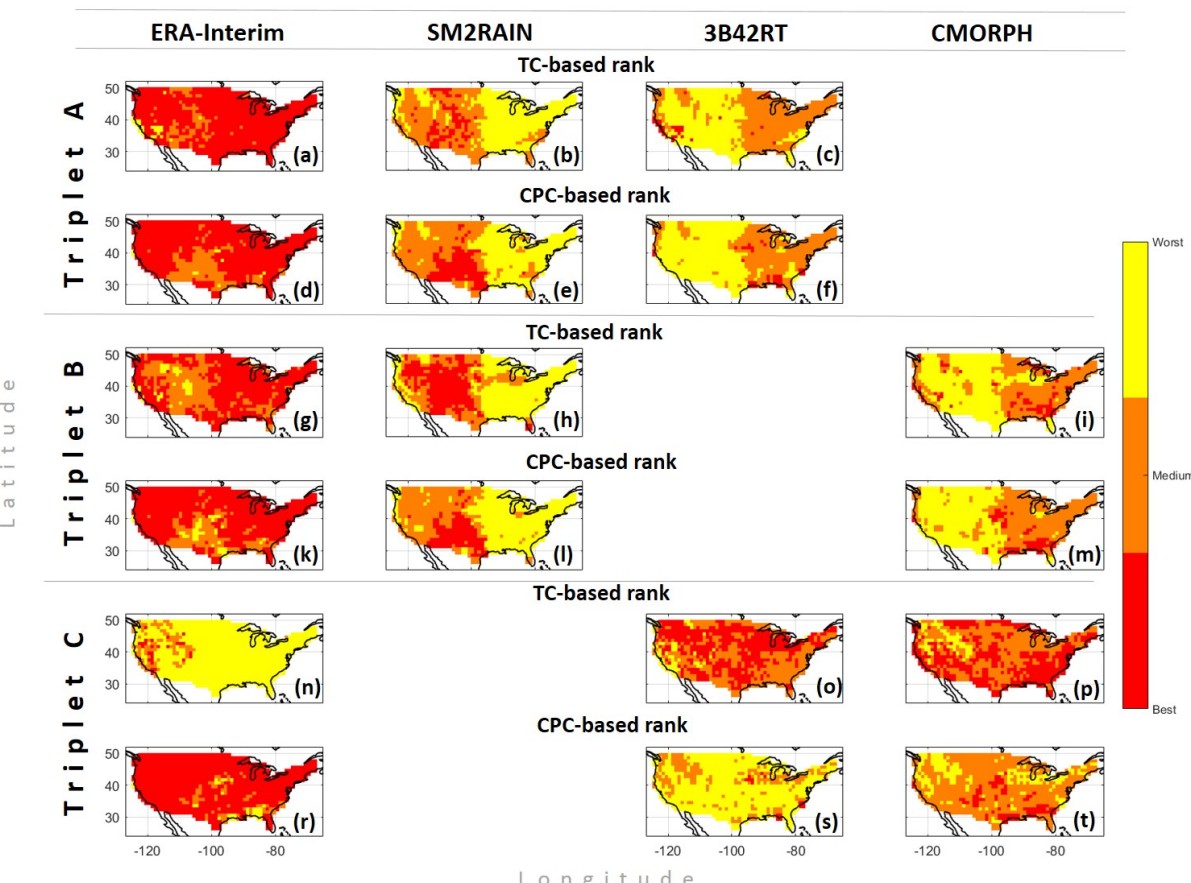

**Figure 3.** Rank based on CPC-based correlation (CPC-based rank in the figure) and TC-based correlation (TC-based rank in the figure) of the triplets: i) ERA-Interim-SM2RAIN-3B42RT (e, f, g), ii) ERA-Interim-SM2RAIN-CMORPH (h, i, l) and iii) ERA-Interim-3B42RT-CMORPH (m, n, o) during the period 2012-2015 using an additive error model.





**Figure 4.** Global correlation of the 3B42RT (a), CMORPH (b), SM2RAIN (c) and ERA-Interim (d) products obtained by Triple Collocation using Triplet A (ERA-Interim-SM2RAIN-3B42RT) for 3B42RT, ERA-Interim and SM2RAIN and Triplet B (ERA-Interim-SM2RAIN-CMORPH) for CMORPH.



| ADDITIVE ERROR | | | | | | |
|---|---|---|---|---|---|---|
| | | **Era-Interim** | **SM2RAIN** | **3B42RT** | **CMORPH** | **CPC** |
| | **Correlation with CPC** | 0.68 | 0.57 | 0.52 | 0.57 | - |
| **Triplet** | **Products** | **Triple Collocation** | | | | |
| A | **ERA - SM2RAIN - 3B42RT** | 0.79 | 0.57 | 0.57 | - | - |
| B | **ERA - SM2RAIN - CMORPH** | 0.73 | 0.63 | - | 0.58 | - |
| C | **ERA - 3B42RT - CMORPH** | 0.43 | - | 0.68 | 0.76 | - |
| D | **ERA - 3B42RT - CPC** | 0.79 | - | 0.57 | - | 0.87 |
| E | **ERA - CMORPH - CPC** | 0.76 | - | - | 0.60 | 0.91 |
| MULTIPLICATIVE ERROR | | | | | | |
| | | **Era-Interim** | **SM2RAIN** | **3B42RT** | **CMORPH** | **CPC** |
| | **Correlation with CPC** | 0.53 | 0.43 | 0.38 | 0.50 | - |
| **Triplet** | **Products** | **Triple Collocation** | | | | |
| A | **ERA - SM2RAIN - 3B42RT** | 0.63 | 0.53 | 0.43 | - | - |
| B | **ERA - SM2RAIN - CMORPH** | 0.68 | 0.55 | - | 0.62 | - |
| C | **ERA - 3B42RT - CMORPH** | 0.43 | - | 0.68 | 0.76 | - |
| D | **ERA - 3B42RT - CPC** | 0.65 | - | 0.42 | - | 0.84 |
| E | **ERA - CMORPH - CPC** | 0.66 | - | | 0.57 | 0.79 |

**Table 1.** Mean CPC-based correaltion, $R$ (correlation with CPC in the table) and TC-based correlations ($\rho$) between different triplets for the additive and for the multiplicative error models. The "Triplet" column refers to the naming convention applied in the text.





| Triplet | Products | SPATIAL CORRELATION | | | |
|---|---|---|---|---|---|
| | | Era-Interim | 3B42RT | CMORPH | SM2RAIN |
| | | Additive error model | | | |
| A | ERA - 3B42RT - SM2RAIN | 0.79 | 0.74 | - | 0.84 |
| B | ERA - CMORPH - SM2RAIN | 0.86 | - | 0.61 | 0.84 |
| C | ERA - 3B42RT - CMORPH | 0.96 | 0.28 | 0.07 | - |
| | | Multiplicative error model | | | |
| A | ERA - 3B42RT - SM2RAIN | 0.380 | 0.751 | - | 0.648 |
| B | ERA - CMORPH - SM2RAIN | 0.265 | - | 0.798 | 0.570 |
| C | ERA - 3B42RT - CMORPH | 0.508 | 0.508 | 0.706 | - |
| | | % RANK IDENTIFIED | | | |
| | | Additive error model | | | |
| A | ERA - 3B42RT - SM2RAIN | 80% | 81% | - | 72% |
| B | ERA - CMORPH - SM2RAIN | 65% | - | 74% | 65% |
| C | ERA - 3B42RT - CMORPH | 6% | 10% | 41% | - |
| | | Multiplicative error model | | | |
| A | ERA - 3B42RT - SM2RAIN | 65% | 71% | - | 60% |
| B | ERA - CMORPH - SM2RAIN | 48% | - | 51% | 67% |
| C | ERA - 3B42RT - CMORPH | 11% | 15% | 50% | - |

**Table 2.** Spatial correlation $SC$ between $\rho_{t,X_i}$ and $R_{X_i}$ and percentage of rank correctly identified obtained with the different triplets considered in the study. The "Triplet" column refers to the naming convention applied in the text.