# Peer review of "An assessment of the performance of global rainfall estimates without ground-based observations"

_Hydrology and Earth System Sciences, 2017_

## Referee Comment (RC1) · Anonymous Referee #1 · 18 Apr 2017

**An assessment of the accuracy of global rainfall estimates without ground-based observations**

C. Massari, W. Crow, L. Brocca

Summary: This study proposes to use a Triple Collocation (TC) method to asses uncertainties associated with rainfall estimates using different products at the daily/1degree scale. The study addresses important issues that are relevant to the HESS readership. I recommend this manuscript for publication after minor revisions.

Page 1:
1. Title and throughout the text: I think there is some confusion regarding the difference between accuracy and precision (uncertainty vs. error, systematic vs. random). Accuracy refers to how close a measured value is to a standard value (i.e., the "true" value). Precision describes the statistical variability. Accuracy and precision are independent. It seems to me that the TC method provides a measure of precision (error variances and correlations), but no information regarding the accuracy, which would require a reference/benchmark. I urge the authors to clarify this difference in the text and revise the text where needed.
2. Line 1: remove "value"
3. Line 4: remove "of this variable"
4. Line 5: replace "among" with "with"
5. Line 18: is it really the best product? It is if precision is the chosen criterion, but it may not be the case if accuracy is considered to be more important.
Page 2:
6. Line 35: replace "provides" with provide"
Page 3:
7. Lines 3 and 6: add "the" before "TC"
8. Line 9: rephrase as "if each product is affected by mutually-independent errors"
9. Line 20: deleted "analysis" and "the"
10. Lines 20-25: CMORPH and TMPA 3B42RT are not completely independent. Can the authors explain what is the implication with using these two products in the TC analysis?

11. Line 28: rephrase as "Section 2 presents datasets and methods;"
12. Line 31: rephrase as "Results and discussion are shown in Section 3 and the final remarks are presented in Section 4."

Page 4:

13. Line 13: delete "a"

14. Lines 2-4: this is a fair model only to model the error for the "hit cases" when both sensor and ground truth are larger than zero or for cumulative rain over a long enough period of time. Otherwise, the multiplicative error model would assign zero anytime the sensor measure a zero. Some explanation is given towards the end of the paragraph, but I believe that this should be discussed when multiplicative error models are introduced. The authors can also refer to Tian et al. 2013: Modeling errors in daily precipitation measurements: Additive or multiplicative?

15. Line 6: shouldn't it be "utilizes"? What's the subject of that verb?

Page 9:

16. Line 16: replace "serve to" with "are used to"

Page 10:

17. Line 6: please replace "assessing" with "to assess"
18. Line 32: please rephrase (2 verbs).

Page 11:

19. Line 9: drop the comma.

Page 12:

20. Lines 1-2: can the authors speculate on why this happens?
21. Line 17: rephrase as "this corroborates what shown by…."

Page 13:

22. Line 2: replace "paper" with "study"

---

## Referee Comment (RC2) · Anonymous Referee #2 · 25 Apr 2017

An assessment of the accuracy of global rainfall estimates without ground-based observations

C. Massari, W. Crow, and L. Brocca

This paper presents a novel approach to estimating surface precipitation using retrieved soil moisture. The authors then apply their soil moisture estimates to understanding the uncertainties in satellite rainfall estimates, and indicate which potential rainfall products perform better in different regions of the CONUS and globally. The applicability to the hydrologic modeling community makes it appropriate for publication in HESS. I recommend publication with minor revisions, many of which deal with adding additional clarification for the reader.

1. Page 3, Line 5: certainly not certainty

2. Page 4, Section 2.1.2: I think that the flow of the paper would be improved by including the description of SM2RAIN with the description of ASCAT (or as a sub-section to it) as opposed to the current arrangement of describing the instrument here and the product several sections later.

3. Page 4-5, Section 2.1.3: Readers familiar with the 3B42 product will recognize that you are using the "Real Time" rather than the "Research" version. In the CMORPH description you mention using the raw version that lacks gauge information, this justification should be included as to why you use 3B42RT as well.

4. Page 5, line 8: SSM/I instruments are operated by the US Department of Defense, not NOAA.

5. Page 5, line 23: 1st

6. Page 6, Line 11: should the second i in the square root also be subscripted?

7. Page 9, Line 19: "use" instead of "have utilize"

8. Page 9, Line 24: You indicate that equation (8) is only valid for liquid precipitation, and in the concluding remarks mention that the SM and combined satellite products are less reliable in cases of frozen precipitation/snow cover/frozen surfaces. Are you using the entire 2012-2015 time period, or only the warm seasons? If you are using the entire period, how are you dealing with the winter months?

9. Page 9, Line 28: Remove the word "values"

10. Page 10, Line 19, "are", not "ae"

11. Page 10, Lines 27-28, and Page 12, Line 3: This may be arguing semantics a bit, but the results don't indicate that not using SM2RAIN yields unreliable results. The results indicate that not adhering to the assumptions of the TC method (specifically with respect to having estimates with uncorrelated errors) produces unreliable results. Table 1 indicates that triplets D and E do just as well without SM2RAIN.

12. Page 10, Line 32: Sentence needs revising

13. Page 11, Lines 15-21: It would be nice to have some context as to why the statistics for the multiplicative error are different from the additive. This comes up a bit later (line 32), but could be more up front.

14. As a general comment, it might be interesting to look at the CMORPH and 3B42 with gauge-adjustment in the global comparison. Presumably this would improve their results in data-rich areas and result in no change in data sparse regions. Comparing triplets using the same product both with and without gauge adjustment might also provide some indication of how much improvement the gauge adjustment provides.

---

## Author Comment (AC1) · 8 May 2017

**Responses to the comments of reviewer #1: An assessment of the accuracy of global rainfall estimates without ground-based observations**

**C. Massari, W. Crow, L. Brocca**

Summary: This study proposes to use a Triple Collocation (TC) method to asses uncertainties associated with rainfall estimates using different products at the daily/1degree scale. The study addresses important issues that are relevant to the HESS readership. I recommend this manuscript for publication after minor revisions.

We thank the reviewer for appreciating the value of the paper and for her/his valuable comments.

**Page 1:**

1. Title and throughout the text: I think there is some confusion regarding the difference between accuracy and precision (uncertainty vs. error, systematic vs. random). Accuracy refers to how close a measured value is to a standard value (i.e., the "true" value). Precision describes the statistical variability. Accuracy and precision are independent. It seems to me that the TC method provides a measure of precision (error variances and correlations), but no information regarding the accuracy, which would require a reference/benchmark. I urge the authors to clarify this difference in the text and revise the text where needed.

R: The reviewer makes a very good point here. If TC is to provide a true assessment of accuracy, it requires a perfectly-calibrated, bias-free scaling target which, of course, is not generally available in data-scarce regions. Instead, we utilize TC to provide an assessment of linear correlation against the "true" rainfall accumulation value. We agree that referring to this as "accuracy" is not correct. However, about the term "precision" can be still misleading since the term precision is used in describing the agreement of a set of results among themselves and is usually expressed in terms of the deviation of a set of results from the arithmetic mean of the set (e.g., standard deviation).

Therefore, to resolve this issue, we will use the word "performance" instead of "accuracy" and clarify early in the paper (at the end of the introduction section, pag.3, lines 23) that "performance" is defined in terms of correlation against "true" rainfall values:

Thanks to the ability of the Extended TC (McColl et al. 2014) to provide the correlation against the "unknown" truth, in this study the assessment of the products will be carried out in terms of correlation against "true" rainfall values. As a result, the word "performance" and "TC results" are here onward referred to this index (additional clarifications are provided in section 2.3).

Based on this strategy, the title will be changed in:

"An assessment of the performance of global rainfall estimates without ground-based observations"

Additional changes will be made in the revised manuscript at lines where the term is not appropriate:

•       Abstract line 14. "Results convey the relatively high accuracy of the satellite rainfall estimates in Eastern North and South America, South Africa, Southern and Eastern Asia, Eastern Australia as well as Southern Europe and complementary performances…." That will be changed in

"Results convey the relatively high performance of the satellite rainfall estimates in Eastern North and South America, South Africa, Southern and Eastern Asia, Eastern Australia as well as Southern Europe and complementary performances…"

•       Pag. 10 line 5 "Assessment of CPC accuracy". The section will be titled "Assessment of the CPC product"

•       Pag. 10 line 29. "… Triplet C leads to an overly-optimistic assessment of their accuracy." will be changed to "Triplet C leads to an overly-optimistic assessment of their performance."

•       Pag. 10 line 32. "It is often important to understand which is the best available rainfall product provides the highest relative accuracy." will be changed to "It is often important to understand which is the best available rainfall product provides the highest relative performance".

•       "particular TC configuration approach to assess the accuracy" will be changed to "particular TC configuration approach to assess the performance".

•       Pag. 14 line 10. "Results convey the relatively high accuracy of daily rainfall accumulations" will changed to "Results convey the relatively high performance of daily rainfall accumulations"

2.    Line 1: remove "value".

We will remove it.

3.    Line 4: remove "of this variable"

We will change the sentence in:

"Recent studies have suggested the use of Triple Collocation (TC) to characterize uncertainties associated with rainfall estimates by using three collocated rainfall products."

4.    Line 5: replace "among" with "with"

We will substitute it.

5.    Line 18: is it really the best product? It is if precision is the chosen criterion, but it may not be the case if accuracy is considered to be more important.

Since we are not talking about the accuracy here the sentence is now adequate. See also the discussion to point 1.

**Page 2:**

6.    Line 35: replace "provides" with provide"

We will replace it.

7. Lines 3 and 6: add "the" before "TC"

We will remove "method" at line 3 and added "the" at line 6.

8. Line 9: rephrase as "if each product is affected by mutually-independent errors"

We think the sentence is correct.

9. Line 20: deleted "analysis" and "the"

We will remove "analysis" term.

10. Lines 20-25: CMORPH and TMPA 3B42RT are not completely independent. Can the authors explain what is the implication with using these two products in the TC analysis?

Cross-correlated error in CMORPH and TMPA 3B42RT have large implications for the application of TC. In fact, these implications provide the central motivation for the paper (particularly our proposed use of a soil moisture-based rainfall accumulation product).

This is an important point for us to convey to the reader. To make sure that it is clear, we will add some clarifying details after line 25 about the possible consequences of applying TC to a triplet containing the two products to facilitate the understanding of this important point.

"Note that, given the number of common sensors shared by CMORPH and TMPA 3B42RT the application of TC to the triplet containing the two products will serve for demonstrating the impossibility to use both of them in the same triplet within the TC analysis. Therefore, additional independent source of rainfall accumulation information are needed."

In addition, note that we have already highlighted this point in the original manuscript at lines 8-13 page 3.

"TC can theoretically provide error and correlations of three products (a triplet) without using ground based observations – provided that each of the three products is afflicted by mutually-independent errors. However, given that state-of-the-art satellite rainfall products use a highly-overlapping set of common sensors for the retrieval of rainfall (see section 2.1 for further details), there is an inherent difficulty in obtaining triplets with mutually-independent errors."

11. Line 28: rephrase as "Section 2 presents datasets and methods;"

The sentence will be rephrased.

12. Line 31: rephrase as "Results and discussion are shown in Section 3 and the final remarks are presented in Section 4."

The sentence will be rephrased.

**Page 4:**

13. Line 13: delete "a"

It will be deleted.

**Page 7**

14. Lines 2-4: this is a fair model only to model the error for the "hit cases" when both sensor and ground truth are larger than zero or for cumulative rain over a long enough period of time. Otherwise, the multiplicative error model would assign zero anytime the sensor measure a zero. Some explanation is given towards the end of the paragraph, but I believe that this should be discussed when multiplicative error models are introduced. The authors can also refer to Tian et al. 2013: Modeling errors in daily precipitation measurements: Additive or multiplicative?

In this section (2.2.1), we discuss thoroughly the multiplicative error assumption along with advantages and limitations when it is used within TC. As a results, we think this is a good place for debating the issue related to presence of zero values in the rainfall time series and the potential problems they can create when the log-transformation is applied.

We will add the reference of Tian et al. (2013).

**Page 8**

15. Line 6: shouldn't it be "utilizes"? What's the subject of that verb?

$\rho_{t,X_i}$ refer the correlations of the products, with i=1,2,3 so we used the plural.

**Page 9:**

16. Line 16: replace "serve to" with "are used to"

We will replace it.

**Page 10:**

17. Line 6: please replace "assessing" with "to assess"

We will replace it.

18. Line 32: please rephrase (2 verbs).

We will rephrase it.

**Page 11:**

19. Line 9: drop the comma.

We will drop it.

**Page 12:**

20. Lines 1-2: can the authors speculate on why this happens?

Some explanations are given at page 11 lines 30-35. In particular, the use of daily rainfall accumulations necessitates the removal of a lot of zero values for applying the log-transformation. This might shorten the dataset too much and decrease the robustness of the TC analysis.

21. Line 17: rephrase as "this corroborates what shown by…."

We will rephrase it.

**Page 13:**

22. Line 2: replace "paper" with "study".

We will rephrase it.

---

## Referee Comment (RC3) · Anonymous Referee #1 · 17 May 2017

All comments have been satisfactorily addressed by the authors.

---

## Author Comment (AC2) · 17 May 2017

**Responses to the comments of reviewer #2: An assessment of the accuracy of global rainfall estimates without ground-based observations**

**C. Massari, W. Crow, and L. Brocca**

This paper presents a novel approach to estimating surface precipitation using retrieved soil moisture. The authors then apply their soil moisture estimates to understanding the uncertainties in satellite rainfall estimates, and indicate which potential rainfall products perform better in different regions of the CONUS and globally. The applicability to the hydrologic modeling community makes it appropriate for publication in HESS. I recommend publication with minor revisions, many of which deal with adding additional clarification for the reader.

We would like to thank the reviewer for appreciating the value of the paper and for her/his valuable comments.

1. Page 3, Line 5: certainly not certainty
We will correct the text.

2. Page 4, Section 2.1.2: I think that the flow of the paper would be improved by including the description of SM2RAIN with the description of ASCAT (or as a sub- section to it) as opposed to the current arrangement of describing the instrument here and the product several sections later.

We thank the reviewer for this suggestion. However, we prefer to separate the methodology section from the data description. SM2RAIN can, in fact, be applied to any type of satellite soil moisture observation, and we would not like to not give the false impression that it can run only with ASCAT soil moisture data.

3. Page 4-5, Section 2.1.3: Readers familiar with the 3B42 product will recognize that you are using the "Real Time" rather than the "Research" version. In the CMORPH description you mention using the raw version that lacks gauge information, this justification should be included as to why you use 3B42RT as well.

We will add the following clarification to the revised paper at line 27 of the Introduction section:

"Note that both 3B42RT and CMORPH (raw version) do not include gauge information in their retrieval algorithms."

4. Page 5, line 8: SSM/I instruments are operated by the US Department of Defense, not NOAA.

We will correct the text.

5. Page 5, line 23: 1$^{st}$

We will correct the text.

6. Page 6, Line 11: should the second i in the square root also be subscripted?

Yes, it should be subscripted because it is referring to a diagonal element of the covariance matrix.

7. Page 9, Line 19: "use" instead of "have utilize"

We will correct the text.

8. Page 9, Line 24: You indicate that equation (8) is only valid for liquid precipitation, and in the concluding remarks mention that the SM and combined satellite products are less reliable in cases of frozen precipitation/snow cover/frozen surfaces. Are you using the entire 2012-2015 time period, or only the warm seasons? If you are using the entire period, how are you dealing with the winter months?

We thank the reviewer for rising this important point. We used the entire period 2012-2015; however, we removed periods of snow cover/frozen soil by masking data where the surface state flag (SSF) of the ASCAT product indicates frozen (SSF=2), temporary melting/water on the surface (SSF=3) or permanent ice (SSF=4). In particular, given that the analysis was carried out at a 1-degree spatial resolution, grid cells were masked if more than 50% of their sub-grid areas consisted of ASCAT observations characterized by a SSF equal to 2, 3 or 4. Moreover, data points where we observed solid precipitation from ERA-Interim were also excluded. The latter, in addition to the consideration of the SSF, helped to reduce the probability of having snowy periods and consider only liquid precipitation. Thus, the results of the paper are not affected by snow.

This will be clarified in section 2.1.2:

"ASCAT points characterized by a surface state flag (SSF) of the ASCAT product that indicates frozen (SSF=2), temporary melting/water on the surface (SSF=3) or permanent ice (SSF=4) were excluded from the analysis."

Page 5 line 30:

"Note that, we considered only liquid precipitation in the analysis. Solid precipitation were

excluded by masking out periods experiencing snowfall (using the "large-scale snowfall" variable of ERA-Interim)."

And at page 9 lines 11:

"Given that the analysis was carried out at a 1-degree spatial resolution, grid cells were masked if more than 50% of their sub-grid areas consisted of ASCAT observations characterized by a SSF equal to 2, 3 or 4."

8. Page 9, Line 28: Remove the word "values"

We will remove it.

9. Page 10, Line 19, "are", not "ae"

We will correct the text.

10.   Page 10, Lines 27-28, and Page 12, Line 3: This may be arguing semantics a bit, but the results don't indicate that not using SM2RAIN yields unreliable results. The results indicate that not adhering to the assumptions of the TC method (specifically with respect to having estimates with uncorrelated errors) produces unreliable results. Table 1 indicates that triplets D and E do just as well without SM2RAIN.

The reviewer is right. The sentence is misleading. We will correct it as:

"This suggests those triplets (where not gauge-based observations are used) not containing SM2RAIN provide unreliable results."

11. Page 10, Line 32: Sentence needs revising

We will modify the sentence to read:

"It is often important to understand which is the best available rainfall product among those available in a specific area"

12.   Page 11, Lines 15-21: It would be nice to have some context as to why the  statistics for the multiplicative error are different from the additive. This comes up a bit later (line 32), but could be more up front.

This will be clarified via new text added after line 20:

"One reason of the lower SC for the multiplicative error model assumption could be the necessary removal of zeros from the time series which potentially creates a different

precipitation signal with reduced sampling power."

13.  As a general comment, it might be interesting to look at the CMORPH and 3B42 with gauge-adjustment in the global comparison. Presumably this would improve their results in data-rich areas and result in no change in data sparse regions.  Comparing triplets using the same product both with and without gauge adjustment might also provide some indication of how much improvement the gauge adjustment provides.

While we agree that this would be an interesting extension, it would require a substantial modification of the existing paper and would entail a substantial departure from the specific goal of this analysis (i.e., to demonstrate that the availability of independent SM2RAIN-based rainfall estimates enables rainfall validation without ground-based observations). However, we fully agree that this suggestion would be a valuable topic for future research.

---

## Referee Comment (RC4) · Anonymous Referee #2 · 19 May 2017

I have no further comments for the authors.
* * *

---

## Author Response (AR1)

**Editor**

Dear Dr. Massari,
Thank you for your interests to HESS. We received two expert reviews on this manuscript. Both reviewers acknowledge the importance of the topic and consider the manuscript well-written. I have read the manuscript myself and concur with reviewers' assessments. I would invite you to revise the manuscript with reviewers' comments in mind. I look forward to reading a revised version of this manuscript.

Regards
Lixin Wang

R: We thank the Editor for his decision. In the following, we provide point-by-point answers (in black) to the referees' comments (in red). We also report the changes made in the manuscript in green indicating also their position.

**Referee 1**

Summary: This study proposes to use a Triple Collocation (TC) method to asses uncertainties associated with rainfall estimates using different products at the daily/1degree scale. The study addresses important issues that are relevant to the HESS readership. I recommend this manuscript for publication after minor revisions.

R: We thank the reviewer for appreciating the value of the paper and for her/his valuable comments.

**Page 1:**

1. Title and throughout the text: I think there is some confusion regarding the difference between accuracy and precision (uncertainty vs. error, systematic vs. random). Accuracy refers to how close a measured value is to a standard value (i.e., the "true" value). Precision describes the statistical variability. Accuracy and precision are independent. It seems to me that the TC method provides a measure of precision (error variances and correlations), but no information regarding the accuracy, which would require a reference/benchmark. I urge the authors to clarify this difference in the text and revise the text where needed.

R: The reviewer makes a very good point here. If TC is to provide a true assessment of accuracy, it requires a perfectly-calibrated, bias-free scaling target which, of course, is not generally available in data-scarce regions. Instead, we utilize TC to provide an assessment of linear correlation against the "true" rainfall accumulation value. We agree that referring to this as "accuracy" is not correct. However, about the term "precision" can be still misleading since the term precision is used in describing the agreement of a set of results among themselves and is usually expressed in terms of the deviation of a set of results from the arithmetic mean of the set (e.g., standard deviation).

Therefore, to resolve this issue, we will use the word "performance" instead of "accuracy" and clarify early in the paper (at the end of the introduction section, pag.3, lines 24-27) that "performance" is defined in terms of correlation against "true" rainfall values:

"Thanks to the ability of TC to provide the correlation against the "unknown" truth (ETC, McColl et al. (2014)), in this study the assessment of the products will be carried out in terms of correlation against "true" rainfall values. As a result, the word "performance" and "TC results" are here onward referred to this index (additional clarifications are provided in section 2.3). "

Based on this strategy, the title has been also changed in:

"An assessment of the performance of global rainfall estimates without ground-based observations"

Additional changes have been made in the revised manuscript at lines (of the original manuscript) where the term resulted not appropriate:

• Abstract line 14. "Results convey the relatively high accuracy of the satellite rainfall estimates in Eastern North and South America, South Africa, Southern and Eastern Asia, Eastern Australia as well as Southern Europe and complementary performances…." That has been changed in

"Results convey the relatively high performance of the satellite rainfall estimates in Eastern North and South America, South Africa, Southern and Eastern Asia, Eastern Australia as well as Southern Europe and complementary performances…"

• Pag. 10 line 5 (now pag. 10 line 20) "Assessment of CPC accuracy". The section has been titled "Assessment of the CPC product"

• Pag. 10 line 29 (now pag. 11 line 15). "… Triplet C leads to an overly-optimistic assessment of their accuracy." has been changed to "Triplet C leads to an overly-optimistic assessment of their performance."

• Pag. 10 line 32 (now pag. 11 line 18). "It is often important to understand which is the best available rainfall product provides the highest relative accuracy." is changed to "It is often important to understand which is the best rainfall product among those available in a specific location".

• Pag. 12 line 7 (now pag 12 line 29) "particular TC configuration approach to assess the accuracy" has been changed to "particular TC configuration approach to assess the performance".

• Pag. 14 line 10 (now pag. 14 line 33). "Results convey the relatively high accuracy of daily rainfall accumulations" has been changed to "Results demonstrate the relatively high performance of daily rainfall accumulations…"

2. Line 1: remove "value".

R: We have removed it.

3. Line 4: remove "of this variable"

R: We have changed the sentence in (now pag 1 line 2):

"Recent studies have suggested the use of Triple Collocation (TC) to characterize uncertainties associated with rainfall estimates by using three collocated rainfall products."

4. Line 5: replace "among" with "with"

R: We have substituted it.

5. Line 18: is it really the best product? It is if precision is the chosen criterion, but it may not be the case if accuracy is considered to be more important.

R: Since we are not talking about the accuracy here the sentence is now adequate. See also the discussion to point 1.

**Page 2:**

6. Line 35: replace "provides" with provide"

R: We have replaced it.

**Page 3:**

7. Lines 3 and 6: add "the" before "TC"

R: We have removed "method" at line 3 and added "the" at line 6.

8. Line 9: rephrase as "if each product is affected by mutually-independent errors"

R: We think the sentence is correct.

9. Line 20: deleted "analysis" and "the"

R: We have removed "analysis" term.

10. Lines 20-25: CMORPH and TMPA 3B42RT are not completely independent. Can the authors explain what is the implication with using these two products in the TC analysis?

R: Cross-correlated error in CMORPH and TMPA 3B42RT have large implications for the application of TC. In fact, these implications provide the central motivation for the paper (particularly our proposed use of a soil moisture-based rainfall accumulation product).

This is an important point for us to convey to the reader. To make sure that it is clear, we have added some clarifying details at lines 32-34 pag. 3 about the possible consequences of applying TC to a triplet containing the two products to facilitate the understanding of this important point.

"Note that, given the number of common sensors shared by CMORPH and TMPA 3B42RT the application of TC to the triplet containing the two products will serve for demonstrating the impossibility to use both of them in the same triplet within the TC analysis."

In addition, note that we have already highlighted this point in the original manuscript at lines 8-13 page 3.

"TC can theoretically provide error and correlations of three products (a triplet) without using ground based observations – provided that each of the three products is afflicted by mutually-independent errors. However, given that state-of-the-art satellite rainfall products use a highly-overlapping set of common sensors for the retrieval of rainfall (see section 2.1 for further details), there is an inherent difficulty in obtaining triplets with mutually-independent errors."

11. Line 28: rephrase as "Section 2 presents datasets and methods;"

R: The sentence has been rephrased.

12. Line 31: rephrase as "Results and discussion are shown in Section 3 and the final remarks are presented in Section 4."

R: The sentence has been rephrased.

**Page 4:**

13. Line 13: delete "a"

R: "a" has been deleted.

**Page 7**

14. Lines 2-4: this is a fair model only to model the error for the "hit cases" when both sensor and ground truth are larger than zero or for cumulative rain over a long enough period of time. Otherwise, the multiplicative error model would assign zero anytime the sensor measure a zero.

Some explanation is given towards the end of the paragraph, but I believe that this should be discussed when multiplicative error models are introduced. The authors can also refer to Tian et al. 2013: Modeling errors in daily precipitation measurements: Additive or multiplicative?

R: In this section (2.2.1), we discuss thoroughly the multiplicative error assumption along with advantages and limitations when it is used within TC. As a results, we think this is a good place for debating the issue related to presence of zero values in the rainfall time series and the potential problems they can create when the log-transformation is applied.

We have added the reference of Tian et al. (2013).

**Page 8**

15. Line 6: shouldn't it be "utilizes"? What's the subject of that verb?

R: $\rho_{t,X_i}$ refer the correlations of the products, with i=1,2,3 so we used the plural.

**Page 9:**

16. Line 16: replace "serve to" with "are used to"

R: We have replaced it.

**Page 10:**

17. Line 6: please replace "assessing" with "to assess"

R: We have replaced it.

18. Line 32: please rephrase (2 verbs).

R: We have rephrased it.

**Page 11:**

19. Line 9: drop the comma.

R: We have dropped it.

**Page 12:**

20. Lines 1-2: can the authors speculate on why this happens?

R: Some explanations are given at page 11 lines 30-35. In particular, the use of daily rainfall accumulations necessitates the removal of a lot of zero values for applying the log-transformation. This might shorten the dataset too much and decrease the robustness of the TC analysis.

21. Line 17: rephrase as "this corroborates what shown by…."

R: We have rephrased as suggested.

**Page 13:**

22. Line 2: replace "paper" with "study".

R: We have replaced it.

**Referee 2**

This paper presents a novel approach to estimating surface precipitation using retrieved soil moisture. The authors then apply their soil moisture estimates to understanding the uncertainties in satellite rainfall estimates, and indicate which potential rainfall products perform better in different regions of the CONUS and globally. The applicability to the hydrologic modeling community makes it appropriate for publication in HESS. I recommend publication with minor revisions, many of which deal with adding additional clarification for the reader.

R: We would like to thank the reviewer for appreciating the value of the paper and for her/his valuable comments.

**Page 3:**

1. Line 5: certainly not certainty

R: We have corrected the text.

**Page 4:**

2. Section 2.1.2: I think that the flow of the paper would be improved by including the description of SM2RAIN with the description of ASCAT (or as a sub- section to it) as opposed to the current arrangement of describing the instrument here and the product several sections later.

R: We thank the reviewer for this suggestion. However, we prefer to separate the methodology section from the data description. SM2RAIN can, in fact, be applied to any type of satellite soil moisture observation, and we would not like to not give the false impression that it can run only with ASCAT soil moisture data.

3. Section 2.1.3: Readers familiar with the 3B42 product will recognize that you are using the "Real Time" rather than the "Research" version. In the CMORPH description you mention using the raw version that lacks gauge information, this justification should be included as to why you use 3B42RT as well.

R: We have added the following clarification to the revised paper at page 3 lines 23-24 of the Introduction section:

"(note that both 3B42RT and CMORPH (raw version) do not include gauge information in their retrieval algorithms)"

**Page 5:**

4. line 8: SSM/I instruments are operated by the US Department of Defense, not NOAA.

R: We have corrected the text.

"It incorporates precipitation estimates derived from the PMW on board of the DMSP 13, 14 & 15

(SSM/I) and NOAA-15, 16, 17 & 18 (AMSU-B) satellites as well as AMSR-E and TMI aboard NASA's Aqua and TRMM spacecraft, respectively."

5. Line 23: 1[st]

R: We have corrected the text.

6. Line 11: should the second i in the square root also be subscripted?

R: Yes, it should be subscripted because it is referring to a diagonal element of the covariance matrix.

**Page 9:**

7. Line 19: "use" instead of "have utilize"

R: We have corrected the text.

8. Line 24: You indicate that equation (8) is only valid for liquid precipitation, and in the concluding remarks mention that the SM and combined satellite products are less reliable in cases of frozen precipitation/snow cover/frozen surfaces. Are you using the entire 2012-2015 time period, or only the warm seasons? If you are using the entire period, how are you dealing with the winter months?

R: We thank the reviewer for rising this important point. We used the entire period 2012-2015; however, we removed periods of snow cover/frozen soil by masking data where the surface state flag (SSF) of the ASCAT product indicates frozen (SSF=2), temporary melting/water on the surface (SSF=3) or permanent ice (SSF=4). In particular, given that the analysis was carried out at a 1-degree spatial resolution, grid cells were masked if more than 50% of their sub-grid areas consisted of ASCAT observations characterized by a SSF equal to 2, 3 or 4. Moreover, data points where we observed solid precipitation from ERA-Interim were also excluded. The latter, in addition to the consideration of the SSF, helped to reduce the probability of having snowy periods and consider only liquid precipitation. Thus, the results of the paper are not affected by snow.

This has been clarified in:

1. section 2.1.2 at lines 24-26:

"Prior to the application of SM2RAIN to ASCAT data, the points characterized by a surface state flag (SSF) of the ASCAT product that indicates frozen (SSF=2), temporary melting/water on the surface (SSF=3) or permanent ice (SSF=4) were excluded from the analysis."

2. Page 6 lines 5-6:

"Note that, we considered only liquid precipitation in the analysis. Solid precipitation were excluded by masking out periods experiencing snowfall (using the "large-scale snowfall" variable of ERA-Interim)."

3. And at page 9 lines 22-24:

"Finally, 1°x1°grid cells were masked if more than 50% of their sub-grid areas consisted of ASCAT observations characterized by a SSF equal to 2, 3 or 4."

9. Line 28: Remove the word "values"

R: We have removed it.

**Page 10:**

10. Line 19, "are", not "ae"

R: We have corrected the text.

11. Lines 27-28, and Page 12, Line 3: This may be arguing semantics a bit, but the results don't indicate that not using SM2RAIN yields unreliable results. The results indicate that not adhering to the assumptions of the TC method (specifically with respect to having estimates with uncorrelated errors) produces unreliable results. Table 1 indicates that triplets D and E do just as well without SM2RAIN.

R: The reviewer is right. The sentence is misleading. We have corrected the text as:

Lines 13-14 page 11

"This suggests those triplets not containing SM2RAIN (or CPC) provide unreliable results."

12. Line 32: Sentence needs revising

R: We will modify the sentence to read (Line 18 page 11):

"It is often important to understand which is the best rainfall product among those available in a specific location."

**Page 11:**

13. Lines 15-21: It would be nice to have some context as to why the statistics for the multiplicative error are different from the additive. This comes up a bit later (line 32), but could be more up front.

R: This has been clarified via new text added at lines 5-8 page 12:

"Here, SC values are generally lower than those obtained by TC (based on an assumed additive error model) likely due to necessity of removing zero rain days which modifies the original precipitation time series and reduces the sample size of TC calculations."

14. As a general comment, it might be interesting to look at the CMORPH and 3B42 with gauge-adjustment in the global comparison. Presumably, this would improve their results in data-rich

areas and result in no change in data sparse regions. Comparing triplets using the same product both with and without gauge adjustment might also provide some indication of how much improvement the gauge adjustment provides.

R: While we agree that this would be an interesting extension, it would require a substantial modification of the existing paper and would entail a substantial departure from the specific goal of this analysis (i.e., to demonstrate that the availability of independent SM2RAIN-based rainfall estimates enables rainfall validation without ground-based observations). However, we fully agree that this suggestion would be a valuable topic for future research

[revised manuscript text omitted]